

# Evaluating the adaptive evolutionary convergence of carnivorous plant taxa through functional genomics

Gregory L. Wheeler and  Bryan C. Carstens

Department of Evolution, Ecology, & Organismal Biology, The Ohio State University, Columbus, OH, United States of America

## ABSTRACT

Carnivorous plants are striking examples of evolutionary convergence, displaying complex and often highly similar adaptations despite lack of shared ancestry. Using available carnivorous plant genomes along with non-carnivorous reference taxa, this study examines the convergence of functional overrepresentation of genes previously implicated in plant carnivory. Gene Ontology (GO) coding was used to quantitatively score functional representation in these taxa, in terms of proportion of carnivory-associated functions relative to all functional sequence. Statistical analysis revealed that, in carnivorous plants as a group, only two of the 24 functions tested showed a signal of substantial overrepresentation. However, when the four carnivorous taxa were analyzed individually, 11 functions were found to be significant in at least one taxon. Though carnivorous plants collectively may show overrepresentation in functions from the predicted set, the specific functions that are overrepresented vary substantially from taxon to taxon. While it is possible that some functions serve a similar practical purpose such that one taxon does not need to utilize both to achieve the same result, it appears that there are multiple approaches for the evolution of carnivorous function in plant genomes. Our approach could be applied to tests of functional convergence in other systems provided on the availability of genomes and annotation data for a group.

Corresponding author
Gregory L. Wheeler,
wheeler.1008@osu.edu,
gwheeler.eb@gmail.com

## INTRODUCTION

Convergent evolution provides some of the strongest support for the theory of evolution through natural selection. In the case of evolutionary convergence, organisms that may have very different evolutionary histories (as measured phylogenetically), are driven by similar selective pressures to a highly similar phenotype (*Losos, 2011*). These selective pressures repeatedly create the same adaptive syndrome—a set of characteristics which come together to allow a specific lifestyle or perform a certain task (*Reich et al., 2003*). In many instances in the past, convergent evolutionary syndromes have confounded taxonomists, who (for example) mistakenly grouped New-World and Old-World vultures (*Seibold & Helbig, 1995*), all marine mammals (*Foote et al., 2015*), and many disparate lineages of microscopic organisms (*Scamardella, 1999*; *Palenik & Haselkorn, 1992*; *Gupta, 2000*), into clades which ultimately proved to be paraphyletic.

While phenotypic features of convergent taxa will appear superficially similar, they are not expected to share genomic similarity due to their evolutionary independence. Large number of possible sequence combinations can result in the same protein (*Storz, 2016*) and potentially large number of protein forms and combinations of multiple proteins that can produce the same effect (*Bork, Sander & Valencia, 1993*; *Doolittle, 1994*), so objectively defining an evolutionary syndrome using genomic data is challenging. One possible solution is to define these syndromes as a set of discrete functions rather than as a set of nucleotide sequences. In this way, convergent syndromes are described in the same way they have evolved—adaptively by function—and can be evaluated as convergent or not based on sequence similarity. Gene Ontology (GO) coding (*Ashburner et al., 2000*) provides an objective system by which to achieve this goal. By designating numerical codes for all possible biological activities and components, ranked hierarchically from general to specific, synonymy of function can easily be measured between even distantly related organisms throughout this text, when a discrete GO term is being referenced, it will be presented in italics, whereas when functions are being referenced in the more general sense, it will be presented in plain text. Using either experimentally determined gene/protein function or sequence similarity to previously identified functions, the activities of individual genes are paired with specific numeric codes. Gene Ontology analyses have been used in other studies to determine the functional components to a variety of traits, adaptations and physiologies of interest, including adaptation to high altitudes (*Qiu et al., 2012*), depth tolerance in deep-sea bacteria (*Vezzi et al., 2005*), and a number of human disorders (*Ahn et al., 2003*; *Holmans et al., 2009*); however, these have identified known genes of interest and then drawn conclusions of function *post hoc*. Rather than assigning the Gene Ontology codes first and subsequently determining the functions of particular interest as has been done previously, we can select functions of expected relevance *a priori* in order to allow for quantitative testing of their adaptive relevance by comparing functions in genomes in species that exhibit a convergent function. To the best of our knowledge, this is a novel approach.

## Plant carnivory

One particularly notable convergent polyphyletic group is that of the carnivorous (alternatively, insectivorous) plants. Carnivorous plant taxa were originally classified as a single group due to their most striking and apparent feature, while disregarding features that would typically be used to define a botanical group (e.g., floral morphology; *Primack, 1987*). Subsequent work has demonstrated that a substantial number of phylogenetically distant plant lineages have evolved a carnivorous lifestyle (*Givnish, 2015*), presumably in response to similar selective pressures. As different lineages (or branches of the same lineage) have approached the process of insect trapping and digestion in different ways, this has in some cases made the defining of a plant as carnivorous or non-carnivorous difficult (*Lloyd, 1934*).

*Givnish et al. (1984)* defines a carnivorous plant as one that fulfills two requirements: it must gain some detectable fitness benefit from animal remains in contact with its surfaces, and it must possess adaptations that facilitate the attraction, capture, or digestion

of these prey animals. By considering only functional attributes, this definition allows a wide range of variability in the evolutionary histories and routes of adaptation of plants that are considered carnivores. Currently, nearly 600 angiosperm species are recognized as carnivorous, representing as many as nine independent origins across five families (*Givnish, 2015*). In addition, investigations into possible carnivorous traits in non-vascular plants such as liverworts are ongoing (*Hess, Frahm & Theisen, 2005*), suggesting that evolutionary shifts in nutrient acquisition strategies are perhaps even more common that currently recognized. The multiple origins and evolutionary convergence demonstrated by radiations such as those in *Nepenthes* and *Sarracenia* indicate that plant carnivory is not phylogenetically constrained; rather, it is likely that these plants are limited by their specific nutrient economics (*Bloom, Chapin & Mooney, 1985*), which allow them to outcompete more typical nutrient acquisition strategies only in specific habitats (*Ellison & Gotelli, 2001*; *Ellison et al., 2003*).

Carnivorous plants occupy habitat where there is little competition for sunlight. Previous studies have shown that, by leaf mass, many carnivorous plants have poor photosynthetic yield (*Ellison & Farnsworth, 2005*; *Ellison, 2006*), a likely consequence of the adaptions of their leaves for the capture of insect prey. Additionally, some carnivorous plants invest photosynthetic carbon in the fluids or secretions utilized for prey capture. In *Drosera*, some 3–6% this carbon, which would otherwise be expended on reproduction or vegetative growth, is used to capture prey (*Adamec, 2002*). As a result of these compromises, carnivorous plants only outcompete other plants in habitat where the resources that they sacrifice as a consequence of the carnivorous lifestyle (carbon, water, sunlight) are plentiful, while the resources they specialize in obtaining (nitrogen, phosphorus) are scarce. These environments are likely to be wet and sunny, with acidic, nutrient-deficient soils (*Givnish et al., 1984*; *Ellison & Gotelli, 2001*).

## Carnivory-associated functions

The most apparent trait of carnivorous plants is their ability to break down prey items using digestive enzymes. As digesting animal tissue is presumably not in the repertoire of ancestral angiosperms, a question of interest is how these enzymes have evolved. In many cases, genes for digestive enzymes are apparent modifications of genes utilized in resistance and correspond to pre-existing pathways related to herbivores and pathogens (*Schulze et al., 2012*; *Fukushima et al., 2017*) or other processes present in most plants. Examples of such enzymes include chitinases, which were modified from anti-fungal and insect herbivore deterrence enzymes (*Hatano & Hamada, 2008*; *Renner & Specht, 2012*), proteases, likely derived from those involved in bacterial resistance (*Mithöfer, 2011*), and lipases, which are involved in metabolizing stored energy (*Seth et al., 2014*). Furthermore, it appears that enzymes with similar functions have evolved convergently in taxa with independent carnivorous origins (*Fukushima et al., 2017*), suggesting that it may not be difficult to evolve into the carnivorous niche. However, digestive enzymes may also be obtained through symbiotic interactions with micro- (*Koopman et al., 2010*; *Caravieri et al., 2014*) or macroorganisms (*Midgley & Stock, 1998*; *Anderson & Midgley, 2003*), suggesting that it may be possible to evolve into the carnivorous niche in part by appropriating the

digestive enzymes of other species. While these plants fit *Givnish et al.*'s (*1984*) definition of carnivores, these digestion-associated genes would not be identifiable in the plant itself and thus would not contribute to functional overrepresentation in genomic analyses.

In addition to modifications or resistance genes or the appropriation of enzymes produced by symbionts, evidence suggests that genes used in nutrient transport are particularly important to the carnivorous lifestyle. Plant genomes possess as many as 10 times the number of peptide transport genes compared to other eukaryotes (*Stacey et al., 2002*), in addition to a wide variety of transport pathways for nitrate and ammonium (*Williams & Miller, 2001*). In carnivorous plants, the relative number of these pathways may be even higher. For example, in a transcriptomic analysis of *Utricularia gibba* L., a carnivorous bladderwort with a minute genome of only 80 megabases, 77 unique sequences corresponding to nitrogen transport were identified (*Ibarra-Laclette et al., 2011*). Modification and specialization has also occurred in transporters for other resources. For plants with traps involving rapid movement such as *Dionaea muscipula* Sol. ex J.Ellis, uptake of prey nutrients may be coupled to a trap's electrical potential (*Scherzer et al., 2013*). Modified pathways for osmolite uptake have been identified in *D. muscipula*, which uses the HKT1-type ion channel to absorb sodium without disrupting the action potential of the trap (*Böhm et al., 2016*). Similar adaptations may benefit less active traps as well, as for example in *Sarracenia flava* L. amino acid uptake is dependent on a potassium ion gradient (*Plummer & Kethley, 1964*).

Genomics represent a new approach to investigate the evolution of novel organismal function. While the origin of novel biological functions and their role in adaptation to new habitats and ecological niches has been an important topic in evolutionary biology since the inception of the field (*Darwin & Darwin, 1888*), we now know that genes may be preferentially duplicated and modified, a common route to increased complexity and the possibility of new structures (*Vandenbussche et al., 2003*) and pathways (*Monson, 2003*). In more extreme cases, a whole-genome duplication event precedes an episode of major adaptive change (*Soltis et al., 2009*), leaving a lineage with thousands of redundant additional genes on which evolutionary processes can act. Gene copies with adaptive value are preferentially retained, while others are silenced and eventually lost (*Adams & Wendel, 2005*). If this general pattern is true of the genes involved in plant carnivory, such genes should be identifiable on the basis of function and would be expected to show a signal of overrepresentation in the genome.

Gene Ontology coding is an essential tool for resolving the issue of relating functionally similar (but non-homologous) genes—by design, genes that differ substantially in ancestry but provide the same function should be assigned the same Gene Ontology code(s). These descriptors are originally assigned based on experimental studies of specific genes in model organisms, which later allows non-experimental assignment using sequence homology; however, as automated annotation must be based on the content of a reference database, known biases in these databases must be considered. For example, studies addressing multiple genes often focus on a specific gene class within a specific organism, resulting in an overemphasis of that class in that organism and its relatives; experiment-based annotations will be far more common for model organisms or those of economic interest; and, as
more sequences are assigned function through extrapolation rather than experimentation, those assignments can be further propagated, progressively increasing the distance from the original experimental basis (*Altenhoff et al., 2012*; *Thomas et al., 2012*). In particular, due to these biases and methods of accurately matching samples to references, there is concern that functional divergence may be missed in cases where divergent sequences remain similar, or conversely, that erroneous function may be assigned when there is substantial divergence from the nearest-matching reference sequence. Despite this, it has been previously shown that known functionally-divergent paralogs also diverged (by 32% on average) in GO codes assigned by automation (*Blanc & Wolfe, 2004*) and that genes typically retain highly similar functions at amino acid identity levels as low as 40% (*Sangar et al., 2007*). Thus, there is reason to believe that identifying function from sequence data should be sufficiently accurate at our desired level of specificity.

### Hypotheses

This study seeks to test for a functional genetic signal of evolutionary convergence at the level of the genome. Specifically, it seeks to test whether or not a convergently evolved functional syndrome (i.e., metabolic pathways of carnivory) will rely on the same functions across lineages (as seen in *Yang et al., 2015*). Three possibilities will be considered. First, organisms sharing this syndrome may not be genomically distinct from others. This is possible if the functional changes required for this syndrome are not substantial at the genome level (e.g., changes based on slight modification of regulatory elements or alternative splicing), or if neutral variation among taxa is so substantial that the changes fall within the range of normal lineages. In this case, no signal should be detected differentiating experimental taxa from control samples (i.e., GO codes matching to expected carnivory-associated functions are not overrepresented). Second, a syndrome may require a specific set of functions at high representational levels in every lineage where it arises. This would be expected if the use of certain molecular machinery were unavoidable for a task, preventing evolution of the syndrome by any other pathways. In this case, it would be expected that a strong signal would be detected for functions across all experimental taxa (i.e., GO codes matching to expected carnivory-associated functions are uniformly overrepresented across carnivorous taxa). Lastly, a syndrome may indeed make use of some functions from a set list each time it arises, but not necessarily the same functions in each case. This would occur where there are several ways to address the same problem. A result where many of the predicted functions show strong signal, but with greatly different findings in each taxon, would support this model (i.e., GO codes matching to different carnivory-associated functions are overrepresented in each carnivorous taxon).

## MATERIALS & METHODS

### Identification of carnivory-associated functions

A literature review was conducted to develop a reference set of functions previously found to be associated with plant carnivory. A topic search was performed on Web of Science in December, 2016 with the following parameters: ("carnivorous plant" OR "insectivorous plant") AND ("gene" OR "genome" OR "transcriptome" OR "protein") AND ("digestion"

**Table 1 Carnivory-associated functions identified via literature review.** Functions were matched to Gene Ontology terms and codes using the AmiGO2 database (*Balsa-Canto et al., 2016*). In cases where multiple GO codes are given, they are equivalent to or deprecated from the best-matching current term. See Table S1 for more information.

| Gene Ontology term | GO code | Gene Ontology term | GO code |
|---|---|---|---|
| *actin filament* | GO:0005884 | *heat shock protein activity* | GO:0042026; GO:0006986; GO:0034620 |
| *alpha-galactosidase activity* | GO:0004557 | *lipase activity* | GO:0016298 |
| *alternative oxidase activity* | GO:0009916 | *lipid transport* | GO:0006869 |
| *ammonium transmembrane transport* | GO:0008519; GO:0072488 | *methylammonium channel activity* | GO:0015264 |
| *aspartic-type endopeptidase activity* | GO:0004190 | *peroxidase activity* | GO:0004601 |
| *ATP:ADP antiporter activity* | GO:0005471 | *phosphatase activity* | GO:0016791 |
| *ATPase activity* | GO:0016887 | *phospholipase activity* | GO:0004620 |
| *beta-galactosidase activity* | GO:0004565 | *polygalacturonase activity* | GO:0004650 |
| *beta-glucanase activity* | GO:0052736 | *polygalacturonase inhibitor activity* | GO:0090353 |
| *chitinase activity* | GO:0004568 | *protein homodimerization activity* | GO:0042803 |
| *cinnamyl-alcohol dehydrogenase activity* | GO:0045551 | *ribonuclease activity* | GO:0004540 |
| *cyclic-nucleotide phosphodiesterase activity* | GO:0004112 | *serine-type carboxypeptidase activity* | GO:0004185 |
| *cysteine-type peptidase activity* | GO:0008234 | *sodium ion transmembrane transporter activity* | GO:0022816 |
| *endonuclease complex* | GO:1905348 | *superoxide dismutase activity* | GO:0004784 |
| *formate dehydrogenase complex* | GO:0009326 | *symplast* | GO:0055044 |
| *fructose-bisphosphate aldolase activity* | GO:0004332 | *thioglucosidase activity* | GO:0019137 |
| *glucosidase complex* | GO:1902687 | *water channel activity* | GO:0015250 |
| *glutathione transferase activity* | GO:0004364 | *xylanase activity* | GO:0097599 |

OR "transport"), producing 21 results. Publications discussing specific genes (*Owen, 1999*; *An, Fukusaki & Kobayashi, 2002*; *Scherzer et al., 2013*; *Böhm et al., 2016*) or overviews of putatively carnivory-associated genes (*Ibarra-Laclette et al., 2011*; *Schulze et al., 2012*; *Rottloff et al., 2016*) in sequenced carnivorous plant taxa were included (details in Table S1). From *Dionaea muscipula,* functions of proteins identified via proteomic analysis of the trap fluid (*Schulze et al., 2012*) were listed, with the addition of genes related to transport that had been specifically targeted by other studies (*Owen, 1999*; *Böhm et al., 2016*). Similarly, in *Nepenthes,* proteomic analysis of trap fluid released a list of functions likely to be associated with plant carnivory (*Rottloff et al., 2016*) with other studies assaying for a specific digestion-associated enzyme and detecting transport activity via traps' glandular symplasts (*An, Fukusaki & Kobayashi, 2002*; *Scherzer et al., 2013*, respectively). In the bladderwort *Utricularia gibba* L., transcriptomic analysis was used to detect statistically increased expression of genes in traps and leaves putatively associated with carnivory (*Ibarra-Laclette et al., 2011*). Gene function terms, as given by these publications, were cross-referenced with the AmiGO2 Gene Ontology Database (*Balsa-Canto et al., 2016*) and matched to discrete GO codes that accurately represent their functions (Table 1). Of 54 total terms selected, 36 final GO codes were matched, with five terms synonymized and combined with matched terms and 13 terms having no appropriate match.

## Taxon sampling

GenBank's list of assessed plant genomes was surveyed for the inclusion of plants historically considered to be carnivorous. Four were available: *Cephalotus follicularis* (*Fukushima et al., 2017*), *Drosera capensis* (*Butts, Bierma & Martin, 2016*), *Genlisea aurea* (*Leushkin et al., 2013*), and *Utricularia gibba* (*Lan et al., 2017*). The carnivorous taxa sampled represent three independent origins of plant carnivory (*Genlisea* and *Utricularia* likely sharing a single origin) in three plant orders (Caryophyllales, Oxalidales, and Lamiales). They also exemplify four different strategies for prey-capture. *Cephalotus* is a pitcher/pitfall trap, using a nectar lure, slippery rim, and downward-facing projections to guide prey into a digestive soup and prevent their escape; this strategy is also seen in *Nepenthes,* most *Sarracenia*, and some carnivorous bromeliads. *Drosera* is a sticky-trap plant, with glandular trichomes on its leaves that secrete both sticky compounds to prevent prey's escape and digestive enzymes to break them down; *Pinguicula* and *Byblis* also use this strategy. *Genlisea* is considered a lobster-pot trap, where prey species are guided to a small, funnel-like opening, through which exit is impossible; *Sarracenia psittacina* and, arguably, *Darlingtonia californica* employ this strategy. Lastly, *Utricularia gibba*, an aquatic carnivorous plant, uses a number of air-filled bladders to capture and digest prey. A trigger hair is stimulated as potential prey investigates the trap, releasing an air bubble contained within; the resulting vacuum pulls the prey inside, and the trap closes behind them. While no other carnivorous taxa possess this specific form, it does share some characteristics (a fast-moving trap activated by the stimulation of a trigger hair) with *Aldrovanda* and *Dionaea muscipula*. The trap characteristics, floral morphology, and overall growth form of the carnivorous taxa included in this study are depicted in Fig. 1.

Non-carnivorous plants were also surveyed in order to establish a control range of "typical" flowering plants. All assessed plant genomes for which Gene Ontology-coded annotations are already available were included: *Arabidopsis thaliana* (*Swarbreck et al., 2008*), *Boea hygrometrica* (*Xiao et al., 2015*), *Glycine soja* (*Kim et al., 2010*; *Qi et al., 2014*), and *Oryza sativa* (*Ohyanagi, 2006*). Note that one of the carnivorous taxa, *Genlisea aurea*, also possessed GO annotations. Lastly, the genomes of the two non-carnivorous plants most closely related to carnivorous taxa were included: *Ocimum tenuiflorum* (*Upadhyay et al., 2015*), closest sequenced relative of *Byblis, Genlisea, Pinguicula,* and *Utricularia*; and *Actinidia chinensis* (*Swarbreck et al., 2008*), closest sequenced relative of *Darlingtonia, Heliamphora, Roridula,* and *Sarracenia. Boea hygrometrica,* included for its available annotations, is also within the order of *Genlisea* and *Utricularia*. The reference-range taxa selected cover five orders (Brassicales, Ericales, Fabales, Lamiales, and Poales), including both Monocots and Eudicots; thus, these samples can be considered a reasonable representation of the diversity and variation of angiosperms as a whole (Fig. 2). While pairwise sampling and analysis of related carnivorous and non-carnivorous taxa would be optimal to explicitly control for phylogenetic effects, this is unfortunately not possible at present due to the lack of sequenced genomes for many plant orders and the scarcity of annotated plant genomes in general. However, we expect our current reference sampling

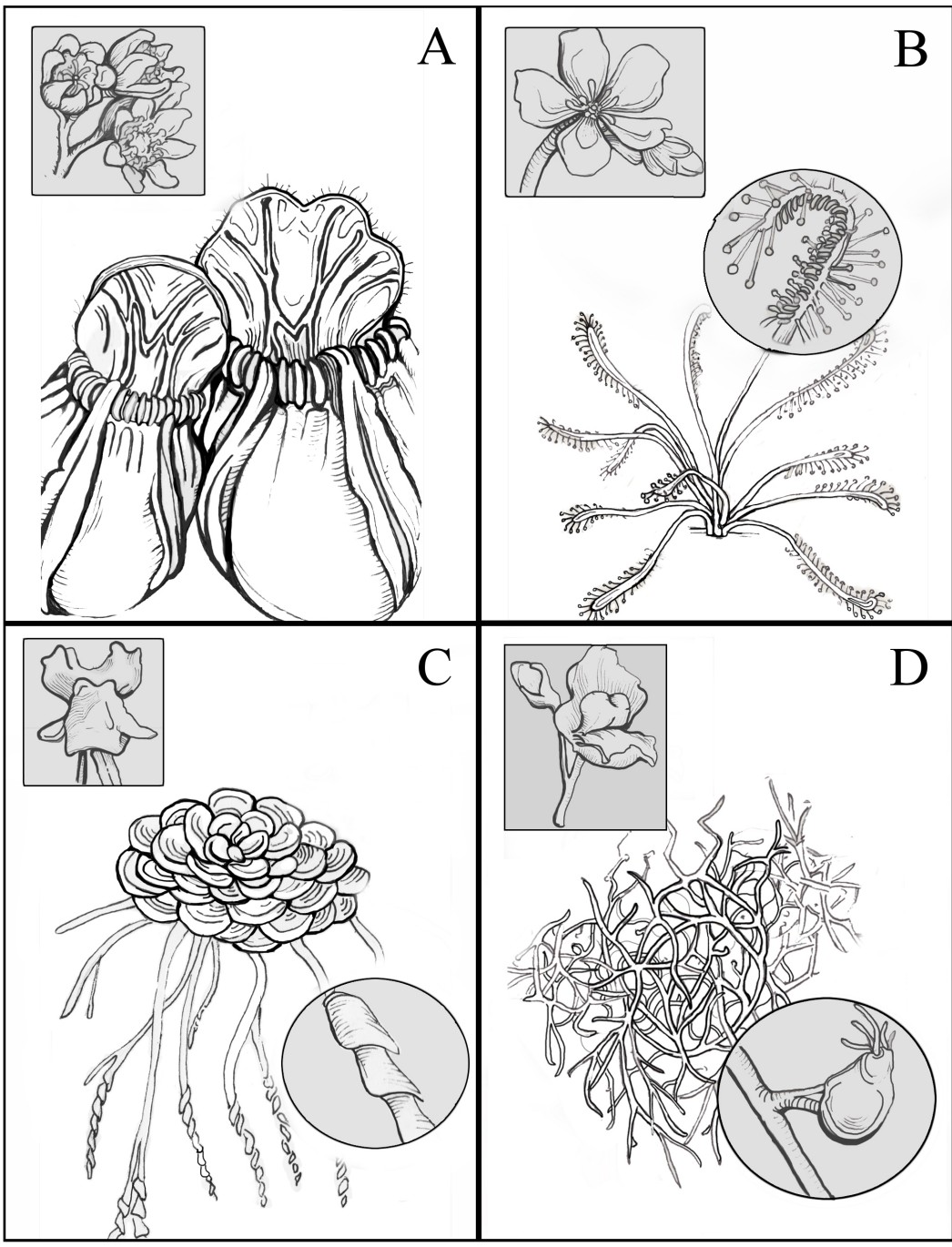

**Figure 1** **Illustrations of the carnivorous taxa included in this study.** Floral characteristics (square inset) and trap morphology (circle inset) are shown, as well as overall growth form. Taxa shown are (A) *Cephalotus follicularis*, (B) *Drosera capensis*, (C) *Genlisea aurea*, and (D) *Utricularia gibba*. Illustrations by Abbie Zimmer, 2017, included with permission.

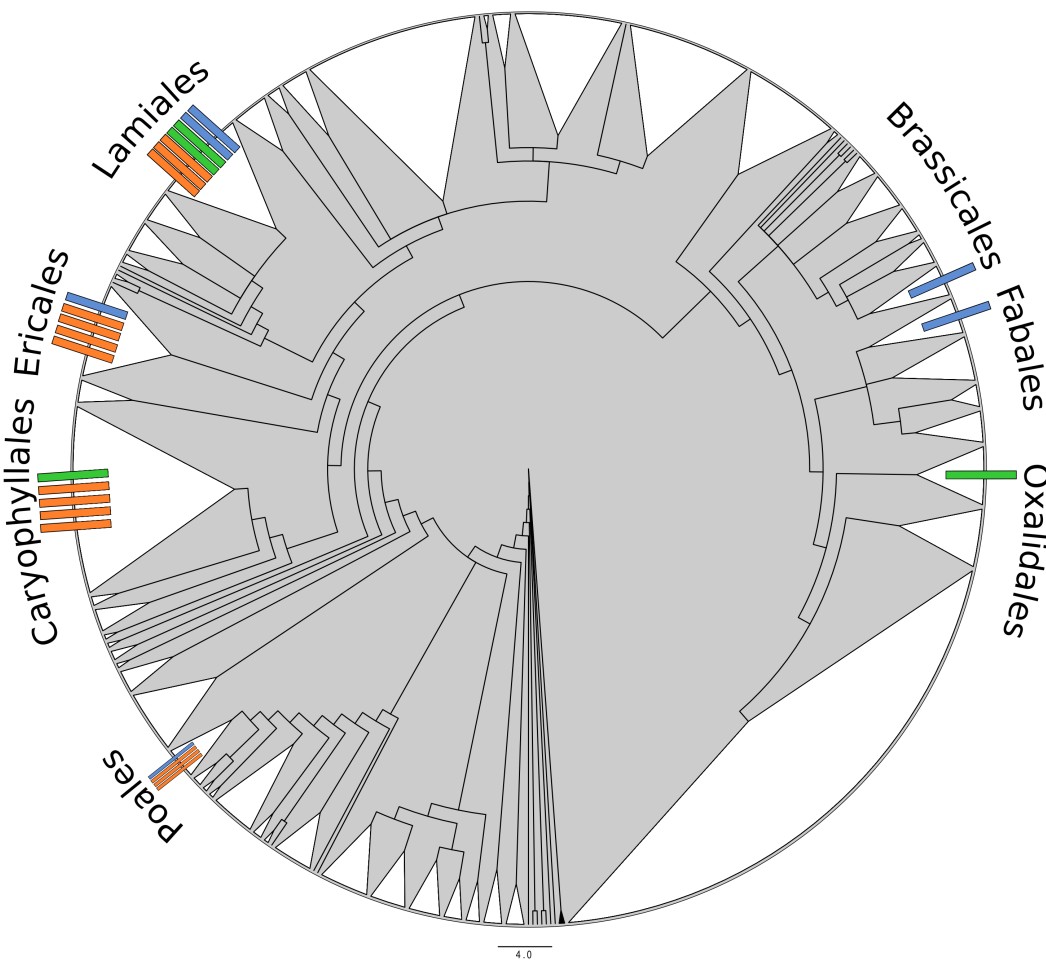

**Figure 2** **Radial phylogeny of all angiosperms, indicating the position of taxa relevant to this study.** White-filled triangles indicate monophyletic plant orders. Each bar indicates one genus. Blue indicates a typical/non-carnivorous control taxon included in this study; green indicates a carnivorous taxon included in this study; orange indicates a carnivorous genus (as listed in *Givnish, 2015*) for which no genome is available. Created from tree data found in *Soltis et al. (2011)*, visualized in FigTree (*Rambaut, 2009*) and manually edited in InkScape.

design, which includes both non-carnivorous representatives from several carnivore-containing orders and a wide phylogenetic range of taxa overall, to somewhat mitigate this potential source of bias.

## Data processing

For taxa lacking GO annotation but having putative genes already identified (e.g., *Cephalotus follicularis*), FASTA-formatted amino acid sequence data was downloaded. The remaining samples (*Actinidia chinensis*, *Ocimum tenuiflorum*, *Drosera capensis*, and *Utricularia gibba*) lacked any usable annotation data. While ideally genes and gene functions are predicted by in-depth transcriptomic studies, the training of species-specific gene identification models, and then confirmed by individual-gene experimental studies, this

is simply unfeasible for studies of diverse sets of non-model taxa. Instead, predictions of genes had to be made on the simpler basis of reading frame detection. Unannotated genomes were downloaded as FASTA-formatted nucleotide sequence and processed with ORFFinder (*Wheeler et al., 2003*) using parameters: (-ml 450 –n false). These parameters were selected to identify putative genes and extract the predicted amino acid sequence. While some error in gene prediction are still likely from this method, parameters were set with the hope of preventing truncated or erroneously-predicted genes from entering the pipeline, e.g., very short of less than 150 amino acids and those contained entirely within the reading frame of another longer gene. Amino acid sequence data was then analyzed via BLAST-P on the Ohio Supercomputer (*Ohio Supercomputer Center, 1987*) with the following parameters: (-db nr -task blastp-fast -seg yes -num_alignments 10 -max_hsps 2 -evalue 1e-3), searching against the non-redundant protein sequence database (*Pruitt, Tatusova & Maglott, 2007*). BLAST outputs were imported into Blast2GO (*Conesa et al., 2005*) and matched to GO codes using the automated ''Mapping'' function. Exported mapping results were then processed via the custom ''AnnotationConverter.pl'' script, to convert data into a more accessible simplified text format. For taxa already accompanied by GO-coded gene annotations (*Arabidopsis thaliana*, *Boea hygrometrica*, *Glycine soja*, *Oryza sativa*, and *Genlisea aurea*), GenBank GBFF files were downloaded. The custom Perl script ''GBFFConverter.pl'' was used to extract genes with associated GO information as simplified text. Using the ''Functionalizer.pl'' Perl script, the resulting text data was then scanned for GO codes matching to the hypothesized carnivory-associated functions selected. Counts of carnivory-associated genes were weighted against total number of genes for which at least one function could be assigned, with the resulting proportions (count of function, per thousand genes) used for subsequent statistical analyses. This process is summarized graphically in Fig. 3. Putative genes that could not be assigned to any function, or that were assigned functions that could not be mapped to any GO codes, were not included in total gene counts or proportional weighting of data. By using a conservative $E$-value parameter in BLAST assignment of protein functions, we hoped to filter out low-certainty annotations, particularly those potentially arising from erroneously-predicted protein sequences. Following this process, a data normalization step was performed to correct for differences in tendency to detect certain functions in BLAST searches vs. from GenBank annotation data.

To correct for differences in the likelihood of assigning a given function between samples accompanied by previous annotation and those coded using BLAST and mapping, *Arabidopsis thaliana* was analyzed by both methods, with the additional data set following the nucleotide sequence data preparation steps detailed above. The raw results of BLAST-annotated data were then multiplied by the quotient of the pre-annotated data results and the *A. thaliana* BLAST data results to produce corrected gene representation data. These data, along with pre-annotated samples that did not require correction, were used in all statistical tests; raw data were subjected to the same analyses, to ensure that the magnitude of changes in results would not be extreme (suggesting the need for more complex methods of error correction). The overall assessment of carnivory-associated function (''Total Carnivorous'' vs. ''None of the Above'') was recalculated for each taxon

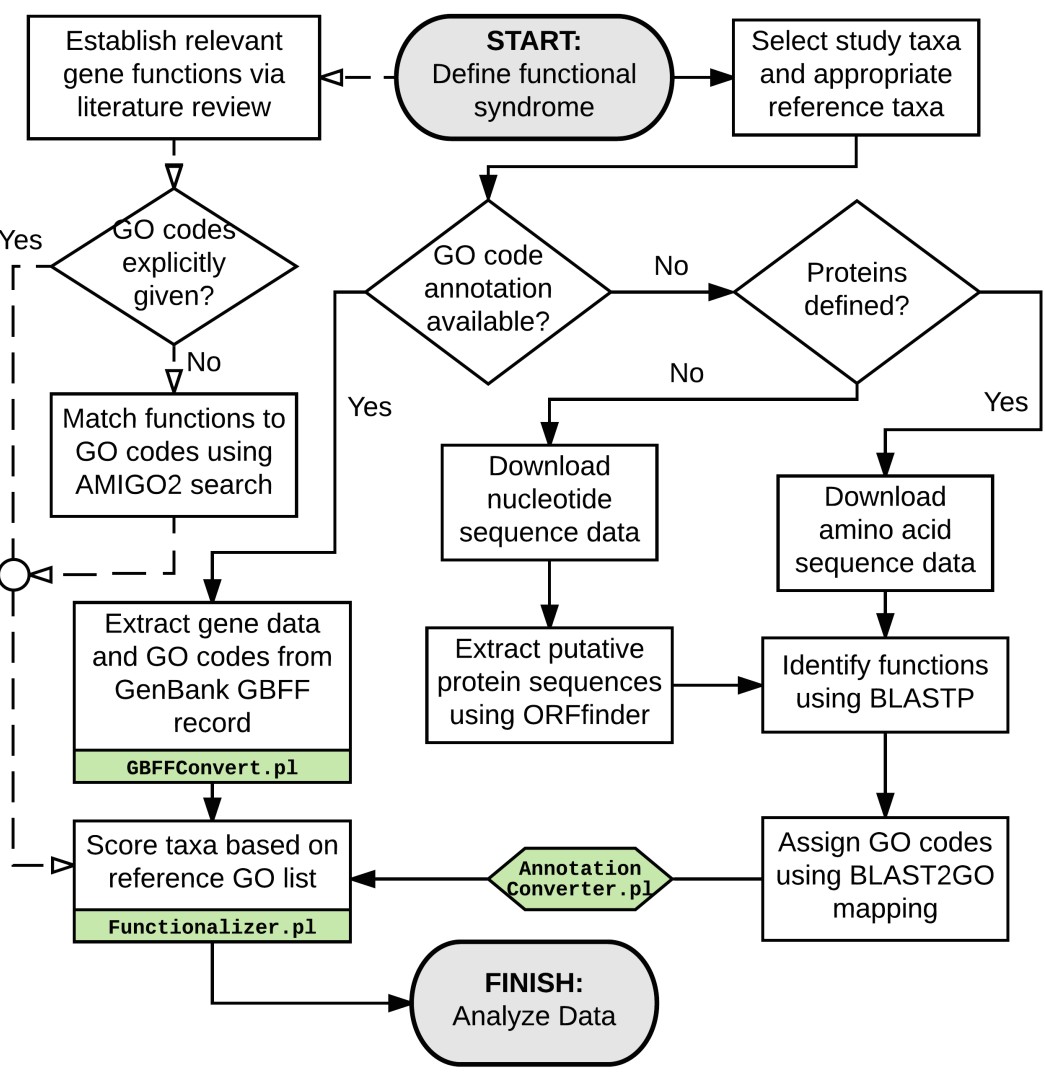

**Figure 3** **Flowchart detailing the preparation and processing steps to obtain gene function represen-
tation data used for subsequent statistical analyses.** Solid lines indicate processing of sampled taxa, while
dashed lines indicate preparation of the reference functional set by which the taxa will be evaluated. Green
boxes indicate stages utilizing custom data-processing scripts (available in Supplemental Information 4).

from the adjusted values of each function and the total gene count ("Total"). Statistical
significance was considered to have six levels ("NS", ".", "*", "**", "***", "****"); number
of levels changed—either increasing or decreasing—were noted. The raw values used in
these corrections are listed in Table S2.

## Statistical analyses

Species were divided into "carnivorous" and "non-carnivorous" groups and analyzed
on 25 criteria (24 carnivory-associated functions, plus the sum representation of all
carnivory-associated functions in the genome) using a series of upper-tailed $t$-tests. To
correct for multiple tests, Storey's correction, which uses a Bayesian approach to determine

**Table 2   General statistics of the plant genomes included in this study.** "Sequence (Mb)" indicates the available genome sequence in million base pairs. "# Genes" indicates the number of putative genes identified, either as indicated in GenBank documentation or detected via ORFfinder. "% Results" indicates the portion of genes that could be associated with at least one GO code. "GO Hits" indicates the total number of GOs matched to a gene across all genes. The number of unique codes present in this number is given as "Unique GOs".

|  | Sequence (Mb) | # Genes | % Results | GO Hits | Unique GOs |
|---|---|---|---|---|---|
| *A. chinensis* | 604.2 | 70,250 | 54.5% | 182,315 | 3,830 |
| *A. thaliana* | 119.7 | 48,350 | 56.3% | 173,184 | 6,503 |
| *B. hygrometrica* | 1,521.3 | 47,778 | 23.2% | 76,891 | 2,916 |
| *G. soja* | 863.6 | 50,399 | 51.4% | 76,044 | 1,421 |
| *O. sativa* | 382.8 | 28,382 | 45.7% | 35,445 | 1,262 |
| *O. tenuiflorum* | 321.9 | 34,920 | 47.8% | 76,891 | 2,916 |
| *C. folicularis* | 1,614.5 | 36,667 | 42.2% | 80,567 | 4,664 |
| *D. capensis* | 263.8 | 89,073 | 24.5% | 113,593 | 3,723 |
| *G. aurea* | 43.3 | 17,685 | 96.6% | 79,194 | 4,883 |
| *U. gibba* | 100.7 | 32,621 | 40.5% | 64,529 | 3,276 |

realistic false discovery rate (FDR) for the numerous tests involved in genome-wide studies (*Storey, 2003*; *Storey & Tibshirani, 2003*; *Dabney, Storey & Warnes, 2010*) was applied, with resulting $q$-values used to assess significance ($\alpha = 0.05$).

Carnivorous taxa were also tested individually, against reference normal distributions created by assessing the values seen in non-carnivorous taxa. Twenty-five reference distributions were used (24 functions + overall), each defined by the median and standard deviation value determined for that function in the non-carnivorous reference taxa. Statistical evaluations were conducted via a series of upper-tailed $Z$-tests, with Storey's correction then used within each series of tests (four sets of 25 tests) to account for repeat testing effects.

# RESULTS

## Genome information

A general assessment of all included genomes for genome size, total gene number, and number of unique Gene Ontology codes identified (a representation of diversity of functions encoded) showed largely overlapping ranges of values (Table 2). Both the largest and smallest genomes analyzed were to carnivorous plants: *Genlisea aurea* at 43.3 Mb and *Cephalotus follicularis* with 1.6 Gb (non-carnivorous plants ranged from 119.7 Mb to 1.5 Gb). Similar results were found for number of genes encoded, ranging from 17,685 in *G. aurea* to 89,073 in *Drosera capensis* (typical plants: 28,382 to 70,250). The largest number of unique GO codes identified was found in *Arabidopsis thaliana*; however, this results from the utilization of *A. thaliana* in the development of plant GO codes. The smallest number was found in *Oryza sativa* (1262), with the largest (after *A. thaliana*) found in *C. follicularis* (4664). Interestingly, when testing for relationships between these factors, no significant ($\alpha = 0.05$ associations were found between genome size and gene number ($p = 0.857$,
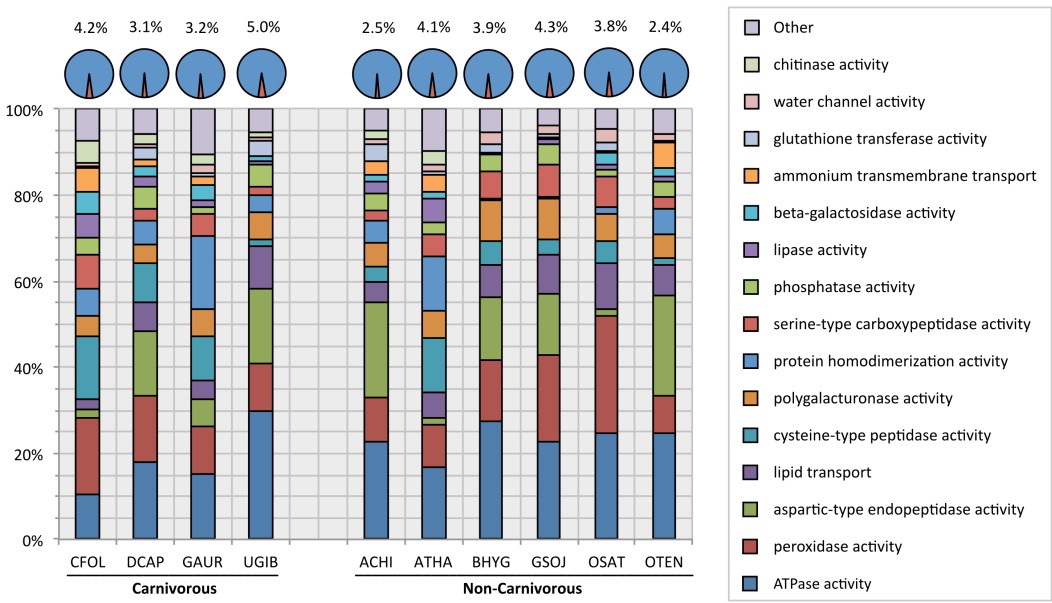

**Figure 4  Chart of proportional representation of carnivorous functions vs. overall gene functions in all taxa sampled.** Pie-charts above indicate total proportion of all carnivorous functions combined (red & percentage) vs. all other genes for which at least one function could be identified (blue). Stacked bars below indicate the proportion ascribed to each carnivory-associated function within the total, sorted from (on average) most represented (bottom) to least represented (top). The final bar, "Other", combines the rarest nine functions, which each on average represent only 0.7% of the carnivory-associated functions detected. A complete numerical view of this data is available in Table S2.

$R^2 = 0.1202$), genome size and number of unique GOs ($p = 0.630$ $R^2 = 0.0909$), or gene number and number of unique GOs ($p = 0.901$, $R^2 = 0.1227$).

When comparing proportion of functionally identifiable genes that could be mapped to a carnivory associated function, there was little difference between carnivorous and non-carnivorous taxa. The percentage of genes mapping to a carnivory associated function in carnivorous taxa ranged from 3.1% to 5.0% of all function-assigned genes; in typical plants, this value ranged from 2.4% to 4.3%. In terms of which specific carnivory associated functions made up each plant's proportion, the representation of each function varied wildly from taxon to taxon (Fig. 4; Table S3).

Statistical comparisons of the genomic representation of each carnivory associated function in carnivorous vs. typical plants yielded two significant ($\alpha = 0.05$ results: "Alternative oxidase activity" ($t = 3.14$, $p = 0.011$, $q = 0.047$) and "ATP:ADP antiporter activity" ($t = 4.00$, $p = 4.30E-03$, $q = 0.037$). A third function, "phospholipase activity" ($t = 2.79$, $p = 0.019$, $q = 0.053$), was detected as significant before correction, but retained only marginal significance ($\alpha = 0.10$ after accounting for multiple tests. Detailed results of these tests are presented in Table 3. For each test, statistical power reaches 50% ($\beta = 0.50$ at an effect size of 1.06 standard deviations and 95% ($\beta = 0.05$ at an effect size of 2.12 standard deviations ($\alpha = 0.05$).
**Table 3  Results of statistical analyses comparing non-carnivorous plants to carnivorous plants for each of 24 carnivory-associated functions, plus the total of all functions.** "*t*" indicates the test statistic of an upper-tailed Student's *t*-test. "*p*" indicates the *p*-value of this test. "*q*" indicates a corrected *p*-value accounting for multiple comparisons, using Storey's correction. Significance ("Sig.") is indicated by bolding and with "*" for $q < 0.05$, "**" for $q < 0.01$, and "***" for $q < 0.001$. A non-bolded "." indicates marginal values ($q < 0.10$), while "NS" indicates non-significance ($q > 0.10$).

| | Z | p | q | Sig. |
|---|---|---|---|---|
| Actin | 0.24 | 0.407 | 0.218 | NS |
| AltOx | **3.14** | **0.011** | **0.047** | * |
| AspPep | 0.01 | 0.496 | 0.241 | NS |
| ATP | −0.24 | 0.590 | 0.241 | NS |
| ATP_ADP | **4.00** | **4.30E−03** | **0.037** | * |
| BGal | 1.90 | 0.062 | 0.127 | NS |
| Chit | −1.82 | 0.944 | 0.324 | NS |
| CinAlc | 0.39 | 0.355 | 0.218 | NS |
| CystPep | −0.32 | 0.619 | 0.241 | NS |
| FrucBPA | 0.66 | 0.266 | 0.217 | NS |
| GlutTrans | 0.29 | 0.391 | 0.218 | NS |
| H2OChan | 1.68 | 0.074 | 0.127 | NS |
| HeatShock | 0.32 | 0.377 | 0.218 | NS |
| Lipase | 0.53 | 0.309 | 0.218 | NS |
| LipTrans | 0.94 | 0.193 | 0.217 | NS |
| NHTrans | 0.64 | 0.278 | 0.217 | NS |
| Perox | −0.14 | 0.552 | 0.241 | NS |
| Phoslip | 2.79 | 0.019 | 0.053 | . |
| Phosp | 0.77 | 0.240 | 0.217 | NS |
| Polygal | −0.75 | 0.763 | 0.284 | NS |
| ProtHomo | 1.27 | 0.122 | 0.174 | NS |
| RiboNuc | −1.10 | 0.841 | 0.300 | NS |
| SerCarPep | −0.29 | 0.608 | 0.241 | NS |
| ThioGluc | −0.18 | 0.570 | 0.241 | NS |
| Total | 0.62 | 0.278 | 0.217 | NS |

Testing of individual carnivorous taxa yielded a total of 13 significant ($\alpha = 0.05$) results and an additional five marginal ($\alpha = 0.10$) results, out of 100 total tests (4 species × 25 distributions). *Genlisea aurea* and *Drosera capensis* had very few functions that showed a signal of genomic overrepresentation. In *Genlisea aurea*, only a single function, "phospholipase activity" ($Z = 2.76$, $p = 2.89E−03$, $q = 0.054$) result reached marginal significance. *Drosera capensis* had one significant function: "alternative oxidase activity" ($Z = 3.72$, $p = 1.01E−04$, $q = 2.45E−03$). *Utricularia gibba* and *Cephalotus follicularis* were found to have a substantial portion of carnivory-associated functions showing strong signals of genomic overrepresentation. In *U. gibba*, five functions reached statistical significance: "alternative oxidase activity" ($Z = 2.36$, $p = 9.17E−03$, $q = 0.025$), "ammonium transmembrane transport" ($Z = 3.58$, $p = 1.74E−04$, $q = 2.09E−03$),

"ATPase activity" ($Z = 3.19$, $p = 7.19\text{E} - 04$, $q = 4.31\text{E} - 03$), "cysteine-type peptidase activity" ($Z = 2.00$, $p = 0.023$, $q = 0.046$), "phosphatase activity" ($Z = 2.65$, $p = 4.05\text{E} - 03$, $q = 0.016$), and "phospholipase activity" ($Z = 2.30$, $p = 0.011$, $q = 0.025$). An additional three test results were marginally significant: "aspartic-type peptidase activity" ($Z = 1.75$, $p = 0.040$, $q = 0.060$), "ATP:ADP antiporter activity" ($Z = 1.54$, $p = 0.062$, $q = 0.083$), and total proportion of carnivory-associated functions ($Z = 1.79$, $p = 0.036$, $q = 0.060$). In *C. follicularis*, seven functions reached significance: "alternative oxidase activity" ($Z = 2.36$, $p = 9.10\text{E} - 03$, $q = 0.030$), "beta-galactosidase activity" ($Z = 4.19$, $p = 1.40\text{E} - 05$, $q = 1.99\text{E} - 04$), "glutathione transferase activity" ($Z = 2.22$, $p = 0.13$, $q = 0.31$), "lipase activity" ($Z = 2.14$, $p = 0.016$, $q = 0.033$), "lipid transport" ($Z = 2.31$, $p = 0.010$, $q = 0.030$), "phospholipase activity" ($Z = 3.50$, $p = 2.28\text{E} - 04$, $q = 0.002$), and "water channel activity" ($Z = 3.08$, $p = 1.04\text{E} - 03$, $q = 4.96\text{E} - 03$). One additional function, "ATP:ADP antiporter activity" ($Z = 1.91$, $p = 0.028$, $q = 0.050$), was marginally significant. Detailed results of these tests are presented in Table 4. For each test, statistical power reaches 50% ($\beta = 0.50$ at an effect size of 2.62 standard deviations and 95% ($\beta = 0.05$ at an effect size of 4.94 standard deviations ($\alpha = 0.05$).

Some changes in results of the above analyses were observed when testing with the raw data sets. In analysis 1 (Table S4), a *t*-test comparison of carnivorous vs. non-carnivorous taxa as groups, *beta-galactosidase activity, phosphatase activity,* protein *homodimerization, thioglucosidase activity,* and *water channel activity* were noted as marginally significant when using uncorrected data; these values dropped below marginal significance (q <0.10) when using corrected data. *Phospholipase* activity was identified as significant ($q < 0.05$) from uncorrected data, but decreased to marginal significance after correction. In analysis 2 (Table S5), changes were as follows: *Genlisea aurea*: no changes. *Drosera capensis*: *ATP:ADP antiporter activity, phosopholipase activity,* and *thioglucosidase activity* declined from significant ($q < 0.05$) to NS ($q > 0.10$), while *cysteine-type peptidase activity* declined from marginal to NS. *Utricularia gibba*: *ammonium transmembrane transport activity* and *ATPase activity* increased from NS to ** ($q < 0.01$), *cysteine-type peptidase activity,* and *phospholipase activity* increased from NS to significant, and *aspartic-type peptidase activity* and total carnivorous function increased from NS to marginal; however, *phosphatase activity* declined from ** to significant, and *cinnamyl-alcohol dehydrogenase activity, polygalactosidase activity,* and *protein homodimerization activity* declined from significant to NS. *Cephalotus follicularis*: *beta-galactosidase activity* increased from NS to *** ($q < 0.001$), *phospholipase activity* and *water channel activity* increased from NS to **, *glutathione transferase activity* increase from marginal to significant, and *lipase activity* and *lipid transferase activity* increased from NS to significant; *heat shock protein activity* decreased from **** ($q < 0.0001$) to NS, *ATP:ADP antiporter activity* declined from ** to marginal, *ATPase activity* and *protein homodimerization activity* decreased from ** to NS, *polygalactosidase activity* declined from significant to NS, and *fructose bisphosphate aldolase activity* declined from marginal to NS.

In total across all tests, 81% of results remained unchanged in designation (Table 5). For those that did change, a decrease in significance was more common (10%) than an

Wheeler and Carstens (2018), *PeerJ*, DOI 10.7717/peerj.4322

**Table 4 Results of statistical analyses comparing non-carnivorous plants to carnivorous plants in four sets, with each evaluating 24 carnivory-associated functions, plus the total of all functions.** "$Z$" indicates the test-statistic of an upper-tailed $Z$-test (equal to number of standard deviations from the mean). "$p$" indicates the $p$-value of this test. "$q$" indicates a corrected $p$-value accounting for multiple comparisons, using Storey's correction. Significance ("Sig.") is indicated by bolding and with "*" for $q < 0.05$, "**" for $q < 0.01$, and "***" for $q < 0.001$. A non-bolded "." indicates marginal values ($q < 0.10$), while "NS" indicates non-significance ($q > 0.10$).

| | Genlisea aurea | | | | Drosera capensis | | | | Utricularia gibba | | | | Cephalotus follicularis | | | |
|---|---|---|---|---|---|---|---|---|---|---|---|---|---|---|---|---|
| | $Z$ | $p$ | $q$ | Sig. | $Z$ | $p$ | $q$ | Sig. | $Z$ | $p$ | $q$ | Sig. | $Z$ | $p$ | $q$ | Sig. |
| Actin | 0.25 | 0.403 | 0.626 | NS | −0.69 | 0.755 | 0.861 | NS | −0.06 | 0.525 | 0.351 | NS | 1.04 | 0.150 | 0.214 | NS |
| AltOx | 0.75 | 0.228 | 0.416 | NS | **3.72** | **1.01E−04** | **2.45E−03** | ** | **2.36** | **9.17E−03** | **0.025** | * | **2.36** | **9.10E−03** | **0.030** | * |
| AspPep | −0.74 | 0.771 | 0.674 | NS | 0.20 | 0.421 | 0.861 | NS | 1.75 | 0.040 | 0.060 | . | −1.17 | 0.880 | 0.544 | NS |
| ATP | −1.50 | 0.933 | 0.697 | NS | −1.22 | 0.889 | 0.861 | NS | **3.19** | **7.19E−04** | **4.31E−03** | ** | −1.67 | 0.952 | 0.544 | NS |
| ATP_ADP | 1.52 | 0.064 | 0.238 | NS | 1.71 | 0.043 | 0.523 | NS | 1.54 | 0.062 | 0.083 | . | 1.91 | 0.028 | 0.050 | . |
| BGal | 1.92 | 0.028 | 0.172 | NS | 0.55 | 0.291 | 0.861 | NS | 0.55 | 0.291 | 0.296 | NS | **4.19** | **1.40E−05** | **1.99E−04** | *** |
| Chit | −0.35 | 0.637 | 0.674 | NS | −1.19 | 0.884 | 0.861 | NS | −0.70 | 0.757 | 0.392 | NS | −1.03 | 0.847 | 0.544 | NS |
| CinAlc | 1.22 | 0.111 | 0.345 | NS | −0.39 | 0.650 | 0.861 | NS | 0.31 | 0.379 | 0.311 | NS | −0.29 | 0.616 | 0.482 | NS |
| CystPep | −1.15 | 0.876 | 0.697 | NS | −0.55 | 0.708 | 0.861 | NS | **2.00** | **0.023** | **0.046** | * | −1.43 | 0.923 | 0.544 | NS |
| FrucBPA | 1.56 | 0.059 | 0.238 | NS | −0.64 | 0.739 | 0.861 | NS | 0.75 | 0.227 | 0.273 | NS | −0.02 | 0.508 | 0.454 | NS |
| GlutTrans | −0.19 | 0.577 | 0.674 | NS | −0.20 | 0.578 | 0.861 | NS | −0.90 | 0.817 | 0.392 | NS | **2.22** | **0.013** | **0.031** | * |
| H2OChan | 0.74 | 0.229 | 0.416 | NS | 0.65 | 0.259 | 0.861 | NS | 0.48 | 0.314 | 0.296 | NS | **3.08** | **1.04E−03** | **4.96E−03** | ** |
| HeatShock | 0.69 | 0.244 | 0.416 | NS | −0.74 | 0.771 | 0.861 | NS | −0.07 | 0.527 | 0.351 | NS | 0.83 | 0.204 | 0.224 | NS |
| Lipase | −0.26 | 0.602 | 0.674 | NS | −0.02 | 0.508 | 0.861 | NS | −0.35 | 0.637 | 0.376 | NS | **2.14** | **0.016** | **0.033** | * |
| LipTrans | 0.69 | 0.245 | 0.416 | NS | 0.41 | 0.340 | 0.861 | NS | −0.65 | 0.741 | 0.392 | NS | **2.31** | **0.010** | **0.030** | * |
| NHTrans | −0.82 | 0.794 | 0.674 | NS | 0.98 | 0.163 | 0.861 | NS | **3.58** | **1.74E−04** | **2.09E−03** | ** | −0.86 | 0.806 | 0.544 | NS |
| Perox | −0.59 | 0.722 | 0.674 | NS | −0.24 | 0.596 | 0.861 | NS | 0.00 | 0.499 | 0.351 | NS | 0.57 | 0.284 | 0.270 | NS |
| Phoslip | 2.76 | 2.89E−03 | 0.054 | . | 0.31 | 0.379 | 0.861 | NS | **2.30** | **0.011** | **0.025** | * | **3.50** | **2.28E−04** | **0.002** | ** |
| Phosp | −1.35 | 0.912 | 0.697 | NS | 0.69 | 0.244 | 0.861 | NS | **2.65** | **4.05E−03** | **0.016** | * | 0.83 | 0.203 | 0.224 | NS |
| Polygal | −0.46 | 0.678 | 0.674 | NS | −1.13 | 0.870 | 0.861 | NS | 0.47 | 0.320 | 0.296 | NS | −0.45 | 0.674 | 0.482 | NS |
| ProtHomo | 2.09 | 0.018 | 0.169 | NS | 0.11 | 0.458 | 0.861 | NS | 0.28 | 0.388 | 0.311 | NS | 0.62 | 0.269 | 0.270 | NS |
| RiboNuc | −0.62 | 0.734 | 0.674 | NS | −0.42 | 0.661 | 0.861 | NS | −0.34 | 0.635 | 0.376 | NS | −0.43 | 0.668 | 0.482 | NS |
| SerCarPep | −0.32 | 0.626 | 0.674 | NS | −0.92 | 0.821 | 0.861 | NS | −0.87 | 0.808 | 0.392 | NS | 1.35 | 0.089 | 0.142 | NS |
| ThioGluc | 0.85 | 0.198 | 0.416 | NS | −0.41 | 0.658 | 0.861 | NS | −0.41 | 0.658 | 0.376 | NS | −0.41 | 0.658 | 0.482 | NS |
| Total | −0.38 | 0.649 | 0.674 | NS | −0.56 | 0.711 | 0.861 | NS | 1.79 | 0.036 | 0.060 | . | 0.86 | 0.195 | 0.224 | NS |

**Table 5** **Effects of data adjustment on statistical significance detected in results.** Change was measured in significance levels, considering six levels: $q > 0.10$ (NS), $q < 0.10$ (.), $q < 0.05$ (*), $q < 0.01$ (**), $q < 0.001$ (***), $q < 0.0001$ (****). "Increase" shows cases where a result went up one or more significance levels; "Decrease" shows cases where a result went down one or more significance levels; "Change > 1" shows cases where a result went either up or down two or more significance levels. "Class" indicates the results based off the categorical correction seen in Table S4 vs. main text Table 3. "Individual" indicates the total of individual comparison results as shown by Table S5 vs. main text Table 4; species names show these comparisons for each species considered separately. "Overall" shows the total of all tests.

| | No change | Increase | Decrease | Change > 1 |
|---|---|---|---|---|
| Class | 19 (76%) | 0 (0%) | 6 (24%) | 0 (0%) |
| Individual | 83 (83%) | 10 (10%) | 5 (5%) | 9 (9%) |
| *C. follicularis* | 17 (68%) | 6 (24%) | 2 (8%) | 6 (24%) |
| *D. capensis* | 21 (80%) | 0 (0%) | 4 (16%) | 2 (8%) |
| *G. aurea* | 25 (100%) | 0 (0%) | 0 (0%) | 0 (0%) |
| *U. gibba* | 20 (80%) | 4 (16%) | 0 (0%) | 1 (4%) |
| Overall | 102 (82%) | 10 (8%) | 12 (10%) | 9 (7%) |

increase (8%). Less than half of all changes (7% of total tests) were of more than a single significance level. Thus, the analyses presented in the main text are performed using the adjusted data.

## DISCUSSION

The analyses presented here were designed to identify similarities in function among carnivorous plants, and we found mixed support for our hypotheses. The null hypothesis ("$H_0$: Carnivorous plant genomes are not distinct from typical plants in functional terms") cannot be rejected for 11 of the 24 functions tested. The first alternate hypothesis ("$H_1$: All carnivorous plants contain a shared functional signal as a result of convergence") is given some support by the results of statistical comparisons between carnivorous and typical plants overall, as it does appear that alternative oxidase activity and ATP:ADP anti-porter activity (as well as, potentially, phospholipase activity) may be commonly overrepresented in carnivorous taxa. Our results support the other alternative hypothesis ("$H_2$: Carnivorous plants are distinct in gene function from typical plants, but this difference varies from taxon to taxon"), as seen in nine (and one additional, marginally) of the 24 functions tested. In short, only a small number of functions appear to be consistently over-represented in taxa sharing the syndrome of plant carnivory; others, while from a predictable set, are over-represented on a taxon-to-taxon basis but in an unpredictable manner. A majority of functions, even if involved in the functional syndrome described, will likely not show a detectable signal, either due to high levels of variation within the control group or because other methods of up-regulation (transcriptional, translational, or structural) have been employed. In any case, our study suggests that plant carnivory can evolve using multiple independent metabolic pathways.

### Overall effects

This study sought to detect a signal of genomic overrepresentation of functions researchers had previously determined were associated with carnivory in plants. The two functions

consistently identified as significantly overrepresented were "alternative oxidase activity" and "ATP:ADP anti-porter activity". Alternative oxidase functions primarily in the mitochondria, as part of the electron transport chain. It is believed to function as a "protective" enzyme, to prevent over-oxidation in the mitochondria and can be activated in response to oxidative stress (*Day & Wiskich, 1995*). For this function to be of common importance to carnivorous taxa, there are three possible explanations: (i) carnivorous plants, due to their digestive function, produce larger amounts of reactive oxygen species (consistent with *Chia et al., 2004*), requiring more alternative oxidase to counteract their negative effects; (ii) alternative oxidase is encoded in a modified form, having been co-opted to perform a different function outside the mitochondrion; or (iii) alternative oxidase is functioning as it would typically, but due to the similar habitat parameters of carnivorous plants, they require its effects more frequently. ATP:ADP anti-porter activity has two functions: It is involved in the maintenance of cellular electrical potential (due to an H+ gradient) in the presence of free fatty acids (*Vianello, Petrussa & Macrì, 1994*) and it allows exchange of cellular ATP for plastid or bacterial-symbiont ADP (*Greub & Raoult, 2003*). In the first case, it may be responsible for interacting with the cellular proton gradient if pH changes substantially during digestion; in the second, it may provide aid to the symbiotic bacteria that assist carnivorous plants in digestion. A third function, "phospholipase activity", was marginally significant. Phospholipases are involved in signaling interactions as well as in metabolism of fatty acids and in degrading cell membranes (*Chapman, 1998*). Carnivorous plants may possess an increased need for complex signaling pathways to regulate their digestive machinery, as well as a clear need to break down cell membranes to access the contents of insect cells.

## Individual taxa

In individual analyses of each carnivorous taxon, alternative oxidase was found to be significant in three species and phospholipase in two species (plus one marginal). Interestingly, ATP:ADP anti-porter activity overall signal was driven by marginal results in two taxa. *Genlisea aurea* and *Drosera capensis* had no significant or marginal functions outside of this set (with one each). In stark contrast, *Utricularia gibba* and *Cephalotus follicularis* both had large numbers of significantly overrepresented carnivory associated functions. Other than alternative oxidase (significant in both), phospholipase (significant in both), and ATP:ADP anti-porter activity (marginal in both), the two taxa did not overlap in any of their other nine (combined) over-represented functions.

*Utricularia gibba* uniquely possessed overrepresentation in *ATPase activity*, *cysteine-type peptidase activity*, *ammonium transmembrane transport*, *phosphatase activity*, and *aspartic-type endopeptidase activity*. Phosphatase, aspartic peptidase, and cysteine peptidase, as catabolic enzymes found localized to the digestive fluids of other carnivorous taxa (*Schulze et al., 2012*; *Rottloff et al., 2016*), most likely have roles in direct digestive function. *Ammonium transmembrane transport*, while required in some amount by all plants, may be more vital for *Utricularia*, which must extract the concentrated nitrogenous products of digestion from an aquatic environment. ATPase in plants is involved in regulation of endocytotic and secretory processes (*Dettmer et al., 2006*), which would logically be

involved in both the release of digestive enzymes and the absorption of digested material. *Utricularia gibba* was the only taxon studied that had even marginal significance in the total genomic proportion of carnivorous functions. Also of note is the vast difference in portion of carnivory-associated functions between *U. gibba* and its close relative *Genlisea aurea*. While both taxa have characteristically-reduced genomes, *G. aurea* has approximately half the genome size and gene number of *U. gibba* (Table 2). It may be that in *Genlisea*, selective pressure strongly favored deletion of duplicated genes, with up-regulation or modification instead occurring at the transcriptional or translational stage.

In *Cephalotus follicularis*, *beta-galactosidase activity*, *water channel activity*, *glutathione transferase activity*, *lipase activity*, and *lipid transferase activity* were found to be uniquely overrepresented. Lipase and beta-galactosidase (which breaks polysaccharide bonds) are likely to have direct involvement in digestion, having also been found in the digestive fluids of other carnivorous taxa (*Schulze et al., 2012*; *Rottloff et al., 2016*); lipid transferase would logically accompany lipase, either to localize lipid substrates or to move the products of their decomposition. As *C. follicularis* must transfer water to the interior of its pitchers for digestive functions to be possible at all, high levels of *water channel activity* is also a logical finding.

**Non-significant functions**

Conversely, 11 functions (*actin*, *chitinase activity*, *cinnamyl-alcohol dehydrogenase activity*, *fructose bisphosphate aldolase activity*, *heat shock protein activity*, *peroxidase activity*, *polygalactonuronase activity*, *protein homodimerization activity*, *ribonuclease activity*, *serine-type carboxypeptidase activity*, and *thioglucosidase activity*) showed no significant over-representation in any taxa sampled. However, due to the relatively low statistical power to detect low to moderate effect sizes with the tests performed, it is possible that these effects do exist but cannot be detected. Even if accepting these negative results as accurate, it is possible that these functions are preferentially utilized in other ways, such as increased transcription, increased protein translation, or increased protein efficiency due to changes in amino acid sequence. Any of these scenarios may also explain why certain functions are overrepresented in the genomes of some carnivorous taxa but not in others.

## CONCLUSIONS

The findings of this study are consistent with expectations of evolutionary convergence. As distant taxa converge on a similar phenotype, predictable functional convergence occurs. This was seen in the cases where there predicted functions, gathered from past studies of carnivorous taxa, were determined to be significantly overrepresented in the taxa sampled. However, this effect was not seen in all functions predicted, nor were the functions showing significant overrepresentation consistent across all four taxa. It is likely that, while these taxa may often show strong signal in some of the functions predicted, the number of potential avenues by which to reach the same practical result is too great for any prediction to hold true in all cases.

The degree of molecular specificity required to meet an organism's needs can also be expected to play a role, with ability to predict a specific functional set increasing

proportional to specificity of the convergent syndrome. In carnivorous plants, a wide range of morphologies (as evidenced by the taxa included in this study) have arisen to reach the same end. In other cases, there is little flexibility in how an organism can reach the needed outcome. For example, organisms that rely on the mimicry of pheromones, such as orchids that imitate bee sex and alarm pheromones, are far less likely to show variation in the functions required for the end-result (*Stökl et al., 2005*; *Stökl et al., 2007*; *Brodmann et al., 2009*). Conversely, even broadly-defined, frequently re-derived evolutionary syndromes may still show repeated selection for specific functional codes. It has been shown that organisms experience substantial convergence of microbiome even for classes as broad as "carnivore" vs. "herbivore" (*Muegge et al., 2011*); it is reasonable to consider that this occurrence may be accompanied by host genome functional convergence as well. However, to detect a signal in these broader groups, where it may be difficult to assemble a manageable list of target syndrome-associated functions, much more thorough sampling would likely be required.

## Future directions

This study is currently limited primarily by the lack of available genomic sequence data for carnivorous taxa, as well as the lack of thorough Gene Ontology annotation of plant taxa in general. This study's BLAST-based annotation methodology is currently impractical for substantially larger taxon sampling, and even in limited taxon sets, greater accuracy is desirable. As more annotated genomes, more consistently high-quality genome assemblies, and more accompanying transcriptomic data sets on which to train gene prediction models become available, it will be possible to more thoroughly assess this phenomenon. As more carnivorous plant taxa are sequenced and annotated (*Nepenthes* and *Dionaea* are expected, as well as *Sarracenia* by the authors), it also becomes possible to refine the reference GO set created for this study, e.g., using functions implicated in previous studies in at least three of 10 taxa. Another potential approach is to apply similar methods to a different functional syndrome. While results may differ based on the evolutionary idiosyncrasies of groups of organisms or from one specific syndrome to another, the same methods could be employed.

## ACKNOWLEDGEMENTS

We thank the members of the Carstens Lab for suggestions and feedback on the methods and presentation of this study. We also thank Abbie Zimmer, who created original illustrations for this study's four carnivorous plant taxa (Fig. 1). Finally, we thank the editor and reviewers for their highly constructive feedback.

### Funding

Gregory L. Wheeler's doctoral studies are supported by a Distinguished University Fellowship, awarded by The Ohio State University Graduate School. Computational resources were provided by the Ohio Supercomputer Center (project PAS1172). The funders had no role in study design, data collection and analysis, decision to publish, or preparation of the manuscript.

### Grant Disclosures

The following grant information was disclosed by the authors:
The Ohio State University Graduate School.
Ohio Supercomputer Center: PAS1172.

### Competing Interests

The authors declare there are no competing interests.

### Author Contributions

- Gregory L. Wheeler conceived and designed the experiments, performed the experiments, analyzed the data, wrote the paper, prepared figures and/or tables, reviewed drafts of the paper.
- Bryan C. Carstens conceived and designed the experiments, contributed reagents/materials/analysis tools, wrote the paper, reviewed drafts of the paper.

### Data Availability

Data and scripts used in this study (included in the Supplemental Files) are also available on GitHub: https://github.com/GWheelerEB/PlantCarnivoryPub-Scripts.

### Supplemental Information

Supplemental information for this article can be found online at http://dx.doi.org/10.7717/peerj.4322#supplemental-information.

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
