# Peer review of "Evaluating the adaptive evolutionary convergence of carnivorous plant taxa through functional genomics"

_PeerJ, doi:10.7717/peerj.4322_

## Round 0.1 · original submission · Major Revisions

This is a very interesting study, and I admire the originality of the approach taken by the authors to test for genomic convergence.

In my view, the points raised by the two reviews are good ones. In addition, I recommend some changes to the manuscript to reassure the reader of the methodological soundness of the work, in roughly descending order of importance.

1. My major concern is about the reliance of this approach on the accuracy, specificity and comprehensiveness of relatively crude gene predictions and GO annotations. For one, I would have expected to see more discussion of some the assumptions and caveats in testing for enrichment using GO (e.g. as in https://doi.org/10.1371/journal.pcbi.1002514 and https://doi.org/10.1371/journal.pcbi.1002386). Some of these would apply even if all the gene predictions themselves were of high quality.

2. I am also concerned by the consequences of having spotty annotations in those genomes where the gene set was derived solely by an untrained ab initio algorithm (ORFFinder). Many genes are likely to be missed or misannotated unless transcriptome and comparative sequence data have been taken into account. I appreciate that obtaining high quality annotations for all the species would not be practical, but the limitations of this should be more clearly spelled out, and it may lead to some artifacts in the results. For instance, could this explain the variation in proportion reported on line 313, due to the difficulty of assigning inaccurately predicted genes to *any* functional category?

3. The potential biases of annotation quality are sufficiently important that I believe the description of how the authors attempt to correct for them belong in the main paper rather than in a Supplementary Note. I would recommend especially that the Methods provide more detail for how 'adjustment factors' were used.

4. I would also recommend that the quality of the gene predictions and annotations be considered along with the number of species and the quality of assemblies in Future Directions on line 462.

(In the authors' favor, I would expect these problems to cause mostly false negative results, so that these tests would at least be conservative)

5. Do all the references in the passage starting on line 68 test for enrichment of GO terms without a priori choice of categories, as seems to be implied? Which if any citations demonstrate that sequence-propagated GO annotations are adequate for finding true/meaningful functional divergence between genomes?

6. I expected a little more discussion of how carnivory might have evolved through genetic changes that would be invisible to this approach (i.e. by means other than through gene duplication), but this is only briefly mentioned (on lines 362 & 428). Do the authors have any reason to expect that gene duplications would be quantitatively more or less important than these other mechanisms?

7. Regarding the section on non-significant functions, on line 421, what is the statistical power of this analysis for detecting significance? If power is low, absence of positive results may not be sufficient evidence to conclude that there were few or no changes to the gene complement contributing to these functions.

8. I may be misreading this, but I believe Table 1 has 19 codes, rather than 20 as reported in the text.

9. Insert into abstract: "the specific functions >that< are overrepresented"

10. line 622, correct journal name.

11. In the figure 2 caption, it took a while for me to understand what "cartooned" meant. Suggestion to use "white-filled" or some other description.

12. Figure 3. The flow implied by the dashed connectors on the left of the diagram is unclear.

13. Figure 4 caption: suggestion to say "percentage" rather than "number", and "combines" rather than "condenses"?

14. Table 1. Do you mean "deprecated" rather than "depreciated"?

Reviewer 1 ·

Basic reporting

The manuscript is clearly written with clear English. Enough background and references are provided. Figures and tables are well described. Although many raw data were provided as supplementary materials, I would suggest the authors to include their custom-made ORF models and GO annotations to ensure the reproducibility.

Experimental design

The question and hypothesis were described well. I found no problem on technical and ethical standard.

Validity of the findings

The current version of the manuscript lacks a statement on their implicit assumption on phylogenetic relationships. Despite of the author’s clear recognition as seen in Fig. 2, the authors did not take into account phylogenetic relationships in their analysis. Their approach implicitly assumes a star tree for analyzed species. In other words, all species were treated as equally distant to each other, without discriminating carnivorous origins (e.g., Utricularia and Genlisea were implicitly treated to have different origins as carnivore). I would suggest the authors to explicitly mention on this problem as a source of potential bias.

Additional comments

The authors reported, by bioinformatics analysis of published genomes, a signature of convergent gene ontology enrichments in separately evolved carnivorous plants. The manuscript may benefit from the following comments:
 L94: In the commentary you cited, I could not find the claim on 16 independent origins. Their description is: “…, these species represent at least nine independent origins of the carnivorous habit per se…”. You might have misunderstood their Table 1, in which genera are labelled with their trap strategies. Some of them share the common carnivorous ancestry even if they use different trap strategies (e.g., three genera in Lentibulariaceae), and therefore carnivorous origins do not total sixteen.
 L95: “Lower plants” might not be appropriate as a terminology. I would suggest “bryophytes” or “non-vascular plants” instead.
 “Neofunctionalization” is not limited to duplicated homeotic regulatory functions. Please rephrase or remove it.
 L157: To my knowledge, GO-based approaches cannot usually identify lineage-specific repurposed genes in genomes which lack experiment-based annotations. Such novel genes would be annotated with GOs identical to their parent genes by the annotation pipelines based on sequence similarity (such as BLAST2GO used in this study). In such cases, they appear to be non-differentiated in function. Indeed, this manuscript did not gain any insights into neofunctionalization at all.
 L371: I failed to understand why you try to associate the GO terms only to digestion rather than to carnivorous syndrome in general.
 L405: In carnivorous plant genomes, not all of enzymes serve as digestive enzymes, so there is no “clear roles in direct digestive function”, at least only by GO enrichment analyses. The same applies to L415.
 Please clearly distinguish GO terms from the rest (e.g., consistently using quotations for GO terms). For example, the statement starting from L417 is quite confusing. It sounds that you actually measured water channel activity, but what you did was the enrichment analysis of the GO term “water channel activity”.

Reviewer 2 ·

Basic reporting

The manuscript #21378 "Evaluating the adaptive evolutionary convergence of carnivorous plant taxa through functional genomics" by Gregory L. Wheeler and Bryan C. Carstens. The presented study shows interesting and important findings which are related with expectations about evolutionary convergence in carnivorous plants. The manuscript is well written in a proper and high-quality language. Also, the authors provide enough references to support the ideas which are addressed in the manuscript. Figures are relevant, high quality, well labeled and well described.

Experimental design

The originality of the research is within aims and scope of the journal. The research question is well defined, relevant and meaningful. Also, methods are well described and the author provides enough information to replicate all analyses performed.

Validity of the findings

The data analysis part of this manuscript has been conducted very thoroughly, even the annotation process from some available genomes was made using a single methodology. This, in my opinion, it's a success because can considerably be reduced the bias that surely exists when genomes which are compared were annotated using distinct methodologies. A Huge effort has also been made by adding statistical support to the analysis.
I conclude saying that I will support this manuscript once that authors address some issues which I summarize in the next section (general comments for the author).

Additional comments

1. To my knowledge, de novo transcriptome assembly in highly heterozygous species typically yields a higher number of contigs than the actual number of genes expressed, thus rendering a redundant and fragmented reference transcriptome. Considering this possibility is not possible to say how many homologs (or orthologs) genes are present in the whole genome using transcriptome data. This is something that the authors do not consider when they referred to some previous publications, e.g. Lines 141 and L 142. "Utricularia gibba L., which has a minute genome only 80 megabases in size, contains at least 77 distinct genes for nitrogen transport functions". This need to be paraphrased or probably change it by something like: "the high number of unigenes (at least 77 distinct transcripts) annotated like nitrogen transporters in de novo transcriptome of Utricularia gibba L. strongly suggest that in the minute genome of this carnivorous plant species, the nitrogen transporters families are expanded in comparison with other plant species.

2. Even when is possible that differences in some gene families could be related with tandem duplication or gene transposition duplication, it is well known that gene duplication and subsequent gene retention or loss (fractionation) are often attributed to recent and/or ancient whole-genome polyploidy events. None mention about these possibilities is presented in this manuscript and in my opinion, this need to be included at least in the discussion section considering that whole-genome duplication and selective contraction, resulting in enormous genetic diversification. It is necessary to discuss if the divergent sub/neo-functionalization events, might have occurred after the whole genome duplication events (which are different for some of the taxon’s which were analyzed). This probably could strengthen the conclusion of the authors in which they suggest that plant carnivory can evolve using multiple independent metabolic pathways.

3. Considering that both U. gibba and G. aurea have significant differences in terms of the genome size and they are members of Lentibulariaceae family, I consider necessary that authors provide an explanation (hypothesis) about why they have substantial differences in terms of the portion of carnivory-associated functions showing strong signals of genomic overrepresentation. This is relevant considering that even when they show different trap system, both carnivorous plant species are considered lack roots and the prey capture is the main source for uptake some nutritional compounds.

---

## Round 0.2 · Minor Revisions

I commend the authors for a thorough response to reviewer's comments, and appreciate the addition to the acknowledgements. The remaining issues are few and easy to address.

1. I do agree that the author's response to Reviewer 1's concern about phylogenetic non-independence is not fully satisfying. The phylogenetic bias cannot be eliminated entirely because of dependence on the sample of genomes that are available, so I suggest *at least* replacing "will account" on line 275 with wording along the lines of "somewhat mitigates".

2. Minor: line 359 - I only count five levels of statistical significance here.

Please see additional minor comments from Reviewer 1.

Reviewer 1 ·

Basic reporting

The new manuscript is improved and answered most of my comments raised earlier. The only problem I found was their new addition of the statement on the phylogenetic dependency.

In their response letter, the authors replied as:
“We attempted to account for potential phylogenetic bias with our sampling design; however, this is never explicitly stated in the text. This was an oversight on our part, and an explanation of our intention with this design has been included to the “Sampling” portion of the Methods.”

This reply and their addition to the main text is less convincing to me. If you attempted to account for the phylogenetic relationships by sampling design without a phylogenetic comparative method, one carnivorous species and one non-carnivore in each lineage should have been sampled so that fair pairwise comparisons can be made (this is a classical workaround to homogenize different phylogenetic distances). Without validation or supporting references for your sampling strategy, one should refrain from stating that the bias was accounted, while the caution of the bias should be clearly stated.

Minor comments:
L106, “such at” might be “such as”.
L177, “functione” might be “function”.
L564, “detected..” -> detected.

Experimental design

no comment

Validity of the findings

no comment

Additional comments

no comment

Reviewer 2 ·

Basic reporting

As I said before, the presented study shows interesting and important findings which provide an interesting study with an original approach to testing for genomic convergence. The manuscript is well written in a proper and high-quality language. Figures are relevant, high quality, well labeled and well described. After one round of review, the author attends the reviewers' comments and provides new references to support the ideas which are addressed in the manuscript.

Experimental design

The hypothesis and aim were clearly described while methods are deep-described with sufficient detail and information to replica the whole analysis.

Validity of the findings

The experimental design provides a novel approach to testing for genomic convergence and the conclusions come from data analysis with enough statistical support.

Additional comments

The authors provide additional information in response to the reviewers' comments. Mostly, all comments were attended and the manuscript was significantly improved in comparison with the previous version. I consider that the manuscript, in the actual format, provides an interesting study with an original approach to testing for genomic convergence. I also believe that the manuscript will be of general interest to the community, mainly for those which employ genomic approaches to understanding problems in evolutionary biology. In my opinion, this manuscript has merits publication in PeerJ.

---

## Round 0.3 · accepted · Accept

The revisions now address all of the substantive issues raised by the reviewers, and I look forward to seeing this "in print".